# Seasonal variation and etiologic inferences of childhood pneumonia and diarrhea mortality in India

Daniel S Farrar[1], Shally Awasthi[2], Shaza A Fadel[1], Rajesh Kumar[3], Anju Sinha[4], Sze Hang Fu[1], Brian Wahl[5], Shaun K Morris[6], Prabhat Jha[1]*

[1]Centre for Global Health Research, St. Michael's Hospital and Dalla Lana School of Public Health, University of Toronto, Ontario, Canada; [2]Department of Pediatrics, King George's Medical University, Lucknow, India; [3]Department of Community Medicine, School of Public Health, Post Graduate Institute of Medical Education and Research, Chandigarh, India; [4]Division of Reproductive Biology, Maternal and Child Health, Indian Council of Medical Research, New Delhi, India; [5]International Vaccine Access Center, Johns Hopkins Bloomberg School of Public Health, Baltimore, United States; [6]Centre for Global Child Health, Division of Infectious Diseases, Hospital for Sick Children and Dalla Lana School of Public Health, University of Toronto, Toronto, Canada

**Abstract** Control of pneumonia and diarrhea mortality in India requires understanding of their etiologies. We combined time series analysis of seasonality, climate region, and clinical syndromes from 243,000 verbal autopsies in the nationally representative Million Death Study. Pneumonia mortality at 1 month-14 years was greatest in January (Rate ratio (RR) 1.66, 99% CI 1.51–1.82; versus the April minimum). Higher RRs at 1–11 months suggested respiratory syncytial virus (RSV) etiology. India's humid subtropical region experienced a unique summer pneumonia mortality. Diarrhea mortality peaked in July (RR 1.66, 1.48–1.85) and January (RR 1.37, 1.23–1.48), while deaths with fever and bloody diarrhea (indicating enteroinvasive bacterial etiology) showed little seasonality. Combining mortality at ages 1–59 months with prevalence surveys, we estimate 40,600 pneumonia deaths from *Streptococcus pneumoniae*, 20,700 from RSV, 12,600 from influenza, and 7200 from *Haemophilus influenzae* type b and 24,700 diarrheal deaths from rotavirus occurred in 2015. Careful mortality studies can elucidate etiologies and inform vaccine introduction.
DOI: https://doi.org/10.7554/eLife.46202.001

*For correspondence:
jhap@smh.ca

## Introduction

Despite substantial declines in the last decade, pneumonia and diarrhea remain the leading causes of mortality in India among children, causing about 190,000 deaths at ages 1–59 months in 2015 (*Fadel et al., 2017*). Indian deaths are about a quarter of global totals from these conditions (*Fadel et al., 2017*; *Liu et al., 2015*). Mortality among children aged 5–14 years is less well studied and recent estimates suggest a combined burden of almost 30,000 pneumonia and diarrhea deaths in India in 2016 (*Fadel et al., 2019*). The declines in pneumonia and diarrhea mortality during the last decade or so have mostly been driven by expanding treatment, improved nutrition, and vaccination against measles and standard childhood antigens (*Bhutta et al., 2013*; *Wong et al., 2019*). Newer vaccines such as rotavirus and pneumococcal conjugate vaccines have been only recently added to India's Universal Immunization Program (*Arora and Swaminathan, 2016*; *Sachdeva, 2017*). Assessing the impact of these newer vaccines requires understanding the baseline mortality and distribution of specific microbiologic etiologies.

Pneumonia and diarrhea exhibit distinct seasonal variation, and pathogen transmission patterns are commonly associated with climatic conditions such as temperature, rainfall, and humidity (*Chadha et al., 2015*; *Chao et al., 2019*; *Fisman, 2012*). Surveillance data from the United States has shown summer diarrhea peaks are associated with bacterial pathogens while winter peaks are associated with viral pathogens (*Glass et al., 2012*). Though similar nationally representative data are lacking in low and middle-income countries, pathogens such as rotavirus exhibit seasonal heterogeneity in India and are most prevalent in cold, dry months (*Kumar et al., 2016*; *Mehendale et al., 2016*).

Here, we describe estimates of pneumonia and diarrhea mortality in Indian children aged 1 month to 14 years from well-structured verbal autopsies in the nationally-representative Million Death Study (MDS; *Aleksandrowicz et al., 2014*; *Jha et al., 2006*), linked with seasonality, climate region-stratified analyses and clinical syndromes. We also combine mortality with prevalence surveys of *Streptococcus pneumoniae*, respiratory syncytial virus (RSV), influenza, *Haemophilus influenzae* type b (Hib), and rotavirus among clinically-confirmed cases of acute respiratory illness and gastroenteritis. We use these data to draw inferences regarding specific microbiologic etiologies and syndromes of these two diseases.

## Results

### Characteristics of subjects

Between 2005–2013, the MDS conducted 243,379 verbal autopsies from nationally representative sampling units for individuals aged 1 month–69 years. Of the 36,336 deaths among children aged 1 month–14 years, two physicians independently attributed 7134 deaths (20%) to pneumonia and 5631 deaths (17%) to diarrhea (*Table 1*). About 57% of pneumonia deaths occurred at ages 1–11 months, while diarrhea deaths occurred more evenly at ages 1–11 months (39%) and ages 1–4 years (43%). Children aged 5–14 years comprised 9% of pneumonia deaths and 18% of diarrhea deaths. Blinded physicians agreed upon a diagnosis of pneumonia or diarrhea in 77% and 86% of cases upon initial double review, respectively. Based on this representative sample, we estimate that 149,000 pneumonia deaths (uncertainty range 114,000–177,000) and 109,000 diarrhea deaths (uncertainty range 92,000–120,000) occurred in children aged 1 month–14 years in India in 2013. Of the 207,043 deaths among adults aged 15–69 years, 28,530 occurred at ages 15–29, 104,093 at ages 30–59, and 74,420 at ages 60–69 years. Of these, dual physicians independently attributed 1637 deaths to pneumonia at ages 15–29 years, 4155 deaths at ages 30–59 years, and 695 deaths at ages 60–69 years (*Table 2*).

### Mortality trends

Childhood pneumonia and diarrhea mortality rates declined between 2005 and 2013 for all ages (*Figure 1*; *Figure 2*; *Figure 1—source data 1*). Among children at ages 1–59 months, pneumonia mortality (per 1000 live births) fell from 5.4 to 3.3 at ages 1–11 months (absolute rate reduction; ARR 39%), and from 3.4 to 2.0 at ages 1–4 years (ARR 41%). Diarrhea mortality fell from 3.0 to 1.8 at ages 1–11 months (ARR 39%), and from 3.5 to 1.8 at ages 1–4 years (ARR 49%). Among children at ages 5–14 years, pneumonia mortality (per 100,000 population) fell from 7.6 to 4.6 (ARR 34%) while diarrhea mortality fell from 14.0 to 7.2 (ARR 44%). Compared to boys, mortality was higher for both conditions in girls for all ages, though steeper declines occurred among girls at ages 1–4 and 5–14 years (e.g. pneumonia ARR 49% for girls aged 1–4 years; pneumonia ARR 33% for boys aged 1–4 years). Compared to urban areas, mortality rates in rural India also fell faster for both diseases except at ages 5–14 years, where diarrhea mortality fell from 8.7 to 2.8 (ARR 66%) in urban areas and from 15.8 to 8.8 (ARR 38%) in rural areas.

### Time series analysis to document seasonality

Childhood pneumonia and diarrhea mortality exhibited significant seasonal variation in India in our time series regression analysis. Among all children, pneumonia mortality peaked twice yearly in July (rate ratio [RR] 1.22, 99% confidence interval [CI] 1.10–1.35) and January (RR 1.66, 99% CI 1.51–1.82), compared to the annual minimum mortality in April (*Figure 3*; *Figure 3—figure supplement 1*). The seasonality of diarrhea deaths differed when stratified by symptom profile. Diarrhea deaths

**Table 1.** Study numbers and weighted percentages of pneumonia and diarrhea deaths recorded in the Million Death Study between 2005 and 2013, among Indian children aged 1 month to 14 years.

| | Pneumonia[1]<br>n (%[3]) | Diarrhea[2]<br>n (%[3]) |
|---|---|---|
| Age | | |
| 1–11 Months | 4203 (57%) | 2291 (39%) |
| 1–4 Years | 2291 (34%) | 2297 (43%) |
| 5–14 Years | 640 (9%) | 1043 (18%) |
| Sex | | |
| Female | 3645 (52%) | 3064 (55%) |
| Male | 3489 (48%) | 2567 (45%) |
| Place of Death | | |
| Home | 5093 (75%) | 4201 (77%) |
| Health Facility | 1694 (20%) | 1112 (17%) |
| Other | 287 (4%) | 264 (5%) |
| Type of Residence | | |
| Rural | 6037 (84%) | 4766 (86%) |
| Urban | 1097 (16%) | 865 (14%) |
| Poorer States[4] | 4352 (76%) | 3709 (81%) |
| Richer States[4] | 2782 (24%) | 1922 (19%) |
| Symptoms | | |
| Fever | 5964 (84%) | 2842 (47%) |
| Cough | 4482 (63%) | 827 (13%) |
| Difficulty breathing | 5715 (81%) | 1876 (32%) |
| Fast breathing[5] | 4229 (61%) | 1085 (19%) |
| Chest indrawing[5] | 3623 (54%) | 728 (12%) |
| Wheezing[5] | 3773 (56%) | 901 (16%) |
| Diarrhea (i.e. loose stools) | 1144 (17%) | 5122 (92%) |
| Blood in stool[6] | 95 (1%) | 787 (13%) |
| Vomiting | 1913 (26%) | 3950 (71%) |
| Abdominal pain | 1102 (15%) | 2291 (43%) |
| Treatment | | |
| Antibiotics for breathing problems[5] | 2946 (41%) | 722 (12%) |
| ORS/other fluids[6] | 628 (9%) | 2963 (53%) |
| Both physicians agreed on initial assignment[7] | 5431 (77%) | 4832 (86%) |
| **Total deaths** | 7134 | 5631 |

[1]We excluded 272 pneumonia deaths with a reported history of measles by the verbal autopsy respondent. [2]We excluded 557 typhoid and paratyphoid fever deaths and 132 deaths with a reported history of measles by the verbal autopsy respondent. [3]We calculated percentages usinga weighted proportion to account for differences in sampling probabilities. [4]Poorer states include the Empowerment Action Group-Assam states of Assam, Bihar, Chhattisgarh, Jharkhand, Madhya Pradesh, Odisha (Orissa before 2011), Rajasthan, Uttarakhand, and Uttar Pradesh. Richer states include all other states and union territories. [5]Only asked of those who reported difficulty breathing (n = 5715 pneumonia deaths, 1876 diarrhea deaths). [6]Only asked of those who reported symptomatic diarrhea (i.e. loose stools; n = 1144 pneumonia deaths, 5122 diarrhea deaths). [7]Physicians were blinded to each other's diagnoses, and agreement is assessed prior to any adjudication or reconciliation of diagnoses.

DOI: https://doi.org/10.7554/eLife.46202.002

**Table 2.** Study numbers and weighted percentages of pneumonia deaths recorded in the Million Death Study between 2005 and 2013, among Indian adults aged 15 to 69 years.

| | Pneumonia[1] n (%[2]) |
|---|---|
| **Age** | |
| 15–29 Years | 1637 (27%) |
| 30–59 Years | 4155 (63%) |
| 60–69 Years | 695 (11%) |
| **Sex** | |
| Female | 3117 (48%) |
| Male | 3370 (52%) |
| **Place of Death** | |
| Home | 5339 (85%) |
| Health Facility | 935 (12%) |
| Other | 90 (1%) |
| **Type of Residence** | |
| Rural | 5082 (75%) |
| Urban | 1405 (25%) |
| Poorer States[3] | 2497 (52%) |
| Richer States[3] | 3990 (48%) |
| Both physicians agreed on initial assignment[4] | 2517 (38%) |
| **Total deaths** | 6487 |

[1]We defined pneumonia deaths using ICD-10 codes A37, H65-H68, H70, H71, J00-J22, J32, J36, J85, J86, P23, or U04. [2]We calculated percentages using a weighted proportion to account for differences in sampling probabilities. [3]Poorer states include the Empowerment Action Group-Assam states of Assam, Bihar, Chhattisgarh, Jharkhand, Madhya Pradesh, Odisha (Orissa before 2011), Rajasthan, Uttarakhand, and Uttar Pradesh. Richer states include all other states and union territories. [4]Physicians were blinded to each other's diagnoses, and agreement is assessed prior to any adjudication or reconciliation of diagnoses.

DOI: https://doi.org/10.7554/eLife.46202.003

without fever and bloody stool showed significant peaks in July (RR 1.66, 99% CI 1.48–1.85) and January (RR 1.37, 99% CI 1.23–1.48), again compared to April. By contrast, mortality from diarrhea cases with both fever and bloody stool showed less and non-significant seasonality (July RR 1.45, 99% CI 1.00–2.08; December RR 1.18, 99% CI 0.81–1.73) and lower overall incidence. Similarly, the few typhoid and paratyphoid fever deaths (which we excluded from our diarrhea case definition) showed no significant seasonality. Adult (ages 15–69 years) pneumonia mortality also demonstrated bimodal peaks though a July peak was not significant at ages 60–69 years (*Figure 3—figure supplement 2*).

Seasonal variation of pneumonia and diarrhea mortality differed substantially by childhood age (*Figure 4*; *Figure 4—figure supplement 1*). Children aged 1–11 months had a higher incidence of pneumonia mortality in December-January (RR 1.72 in December, 99% CI 1.53–1.95) as well as a non-significant peak in July (RR 1.12, 99% CI 0.98–1.28). Significant bimodal peaks were evident at ages 1–4 years (August RR 1.42, 99% CI 1.24–1.63; January RR 1.59, 99% CI 1.40–1.79), but less so at ages 5–14 years (August RR 1.24, 99% CI 1.08–1.41; January RR 1.52, 99% CI 1.35–1.72). Similar to the national pattern, bimodal July–January peaks were present for diarrhea cases lacking fever and bloody stool in all child age subsets except at ages 5–14 years where only a July peak was evident (July RR 1.84, 99% CI 1.55–2.20; January RR 1.01, 99% CI 0.85–1.20). We had too few deaths of diarrhea with fever and bloody stool to stratify by age group.

Climate dynamics were associated with differential seasonal patterns of pneumonia mortality and uniform patterns of diarrhea mortality. Our analysis stratified deaths by Köppen-Geiger climate region – including the hot semi-arid (classes Bsh and Bwh), humid subtropical (classes Cwa and Cwb), and tropical savannah regions (classes Am and Aw; *Figure 5*). Among all children, we

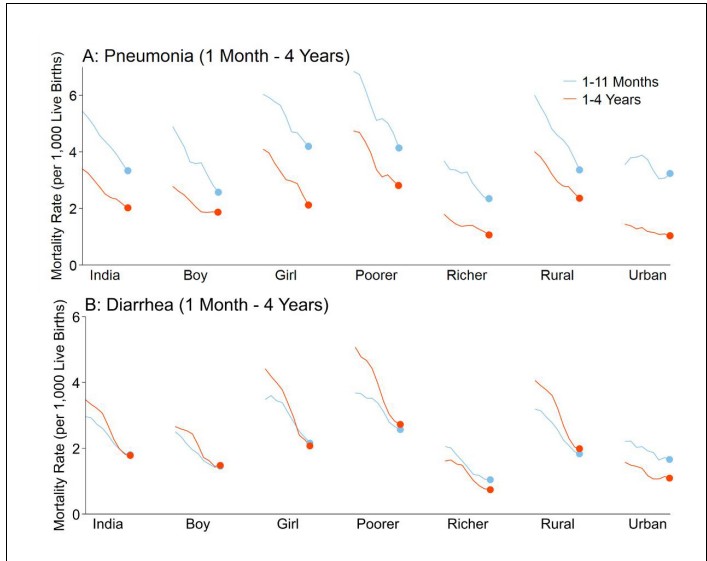

**Figure 1.** National mortality rates of (**A**) pneumonia and (**B**) diarrhea by sex, poorer/richer state, and residence type among Indian children aged 1 month to 4 years between 2005 and 2013. Each x-axis represents 2005–2013. We used a three-year moving average of the weighted proportion of deaths to calculate mortality rates (per 1000 live births). We adjusted death data to reflect 2015 data from the United Nations Population Division and Inter-agency Group for Child Mortality Estimation. Poorer states include the Empowerment Action Group-Assam states of Assam, Bihar, Chhattisgarh, Jharkhand, Madhya Pradesh, Odisha (Orissa before 2011), Rajasthan, Uttarakhand, and Uttar Pradesh. Richer states include all other states and union territories.

DOI: https://doi.org/10.7554/eLife.46202.004

The following source data is available for figure 1:

**Source data 1.** National mortality rates and rate reductions of pneumonia and diarrhea by population subset.
DOI: https://doi.org/10.7554/eLife.46202.005

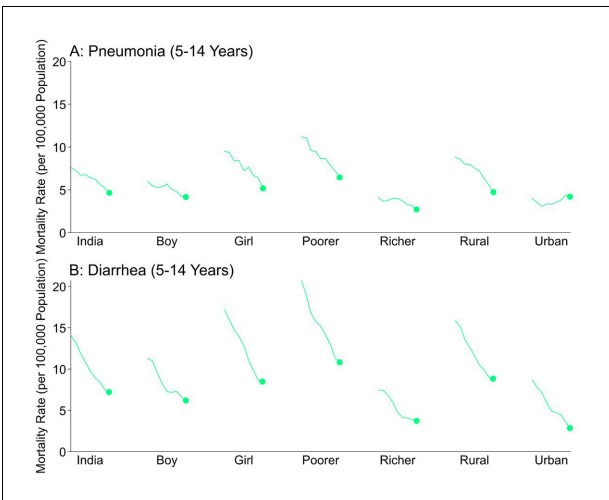

**Figure 2.** National mortality rates of (**A**) pneumonia and (**B**) diarrhea by sex, poorer/richer state, and residence type among Indian children aged 5 to 14 years between 2005 and 2013. Each x-axis represents 2005–2013. We used a three-year moving average of the weighted proportion of deaths to calculate mortality rates (per 100,000 population). We adjusted death data to reflect 2015 data from the United Nations Population Division and Inter-agency Group for Child Mortality Estimation. Poorer states include the Empowerment Action Group-Assam states of Assam, Bihar, Chhattisgarh, Jharkhand, Madhya Pradesh, Odisha (Orissa before 2011), Rajasthan, Uttarakhand, and Uttar Pradesh. Richer states include all other states and union territories.

DOI: https://doi.org/10.7554/eLife.46202.006

observed that the bimodal national pattern occurred only in the humid subtropical region, which exhibited significant peaks in July (RR 1.39, 99% CI 1.16–1.76) and January (RR 1.91, 99% CI 1.58–2.31) (*Figure 6*; *Figure 6—figure supplement 1*). Only unimodal peaks occurred in the tropical savannah (November RR 1.39, 99% CI 1.17–1.67) and hot semi-arid (January RR 1.75, 99% CI 1.39–2.09) regions. Late-year pneumonia peaks were also asynchronous across climate regions, with the tropical savannah region peak occurring earlier in the year. Significant bimodal pneumonia peaks occurred across all climate regions among adults aged 30–59 years but July peaks were not significant at age 15–29 years. Childhood pneumonia deaths occurring between June and August were heavily clustered within the humid subtropical region. By contrast adult pneumonia deaths during the same months were dispersed more evenly across climate regions (*Figure 7*). Seasonal variation of child diarrhea mortality by climate region resembled the bimodal national peaks (*Figure 8*).

## Estimation of etiology-specific mortality

We estimated mortality from five microbiologic etiologies and two internally-defined syndromes by applying etiologic fractions to published estimated deaths among children at ages 1–59 months for 2015 (*Fadel et al., 2017*). We could not retrieve sufficient published data to estimate etiologic fractions for children at ages 5–14 years. Of the 108,000 estimated pneumonia deaths occurring nationally in 2015, we estimated 40,600 deaths (uncertainty range 28,800–42,400) were attributable to *Streptococcus pneumoniae* infection, 20,700 (uncertainty range 13,600–29,000) to RSV, 12,600 (uncertainty range 7,200–20,600) to influenza (A [seasonal and pandemic] and B), and 7200 (uncertainty range 5,000–9,400) to Hib (*Table 3*). We also estimated an excess of 3500 deaths at ages 1–59 months from pneumonia in the humid subtropical region between April and September, compared to the expected number of deaths in the absence of mortality changes between these two annual lows. Of the 82,000 diarrhea deaths in 2015, we estimated 24,700 deaths (uncertainty range 17,200–32,800) were attributable to rotavirus infection and 6200 to the syndrome exhibiting fever and bloody stool.

Substantial heterogeneity in etiologic fractions was present at the administrative region level for all microbiologic etiologies and syndromes except *Streptococcus pneumoniae*, which accounted for roughly 40% of 1–59 month pneumonia mortality across all regions. RSV was most variable between regions, with the etiologic fraction in the South region 3.2 times that in the Northeast (35% [23–47] to 11% [8–15]; *Table 4*; *Table 4—source data 1*). Etiologic fractions of RSV were also negatively correlated with regional pneumonia mortality rates (r = −0.66). The etiologic fraction for influenza was greatest in the Central region (16% [11–21]) and lowest in the Northeast and East regions (6% [<1–17]; *Table 4—source data 2*). Hib pneumonia also had the highest etiologic fraction in the Central region (9% [6–12]), three times greater than the North (3% [2–4]). Heterogeneity was also present in diarrhea etiologic fractions, with 46% (37–56) of diarrhea mortality at 1–59 months in the North region attributable to rotavirus, 2.1 times more than in the Central region (22% [16–27]; *Table 4—source data 3*). Finally, the etiologic fraction of diarrhea with fever and bloody stool was three times greater in the Northeast (12%; no range) than in the South (4%; no range), and etiologic fractions were positively associated with regional diarrhea mortality rates (r = 0.73). No other specific patterns of infection were strongly associated with regional mortality rates, though overall regional pneumonia and diarrhea mortality rates were strongly correlated (r = 0.97).

## Discussion

This study is the first to document the known etiologies of pneumonia and diarrhea using standardized, nationally representative physician-coded verbal autopsy data in India, the largest contributor to global child pneumonia and diarrhea mortality. Nationally, pneumonia and diarrhea both demonstrated bimodal seasonal variation and peaked in July and January. Pneumonia mortality rates were highest at ages 1–11 months, particularly during the cooler months of December and January. Diarrhea deaths with fever and bloody stool, indicative of enteroinvasive bacterial diarrhea and microbiologic etiologies such as *Shigella spp.* showed little seasonality. Climatic factors including temperature and rainfall drove regional differences in pneumonia mortality, including an excess 3500 deaths between June and August in India's humid subtropical region (including most of the major states of Uttar Pradesh and Bihar). These same regions also showed marked geographic clustering of pneumonia deaths in children, whereas adult pneumonia deaths were dispersed. Verbal

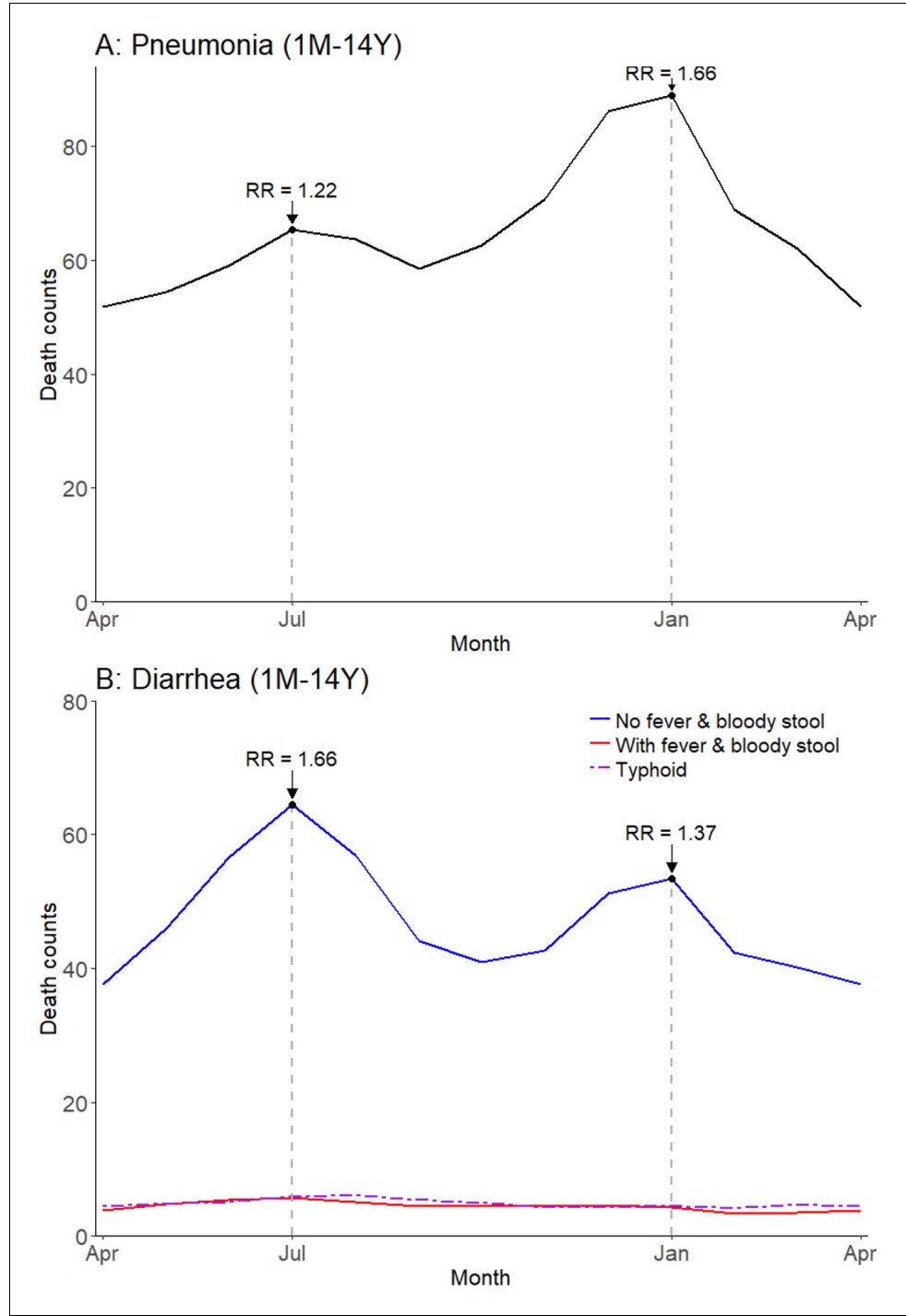

**Figure 3.** Average seasonal patterns of pneumonia (**A**) and diarrhea (**B**). Diarrhea deaths are split into subsets of cases with/without both fever and bloody stool, as well as typhoid and paratyphoid fever deaths. We defined pneumonia deaths using ICD-10 codes A37, H65-H68, H70, H71, J00-J22, J32, J36, J85, J86, P23, or U04. We defined diarrhea deaths using ICD-10 codes A00, A02-A09, and distinguished further based on symptoms reported in the VA (deaths exhibiting both fever and bloody stool; deaths not exhibiting both fever and bloody stool). We defined typhoid and paratyphoid fever deaths using ICD-10 code A01. Each horizontal axis represents an average yearly span from April to April. We determined seasonal patterns using monthly counts of death and modeled using Poisson regression. Rate ratios (RR) are calculated within each disease and are compared to annual minimum mortality in the month of April.

DOI: https://doi.org/10.7554/eLife.46202.007

*Figure 3 continued on next page*

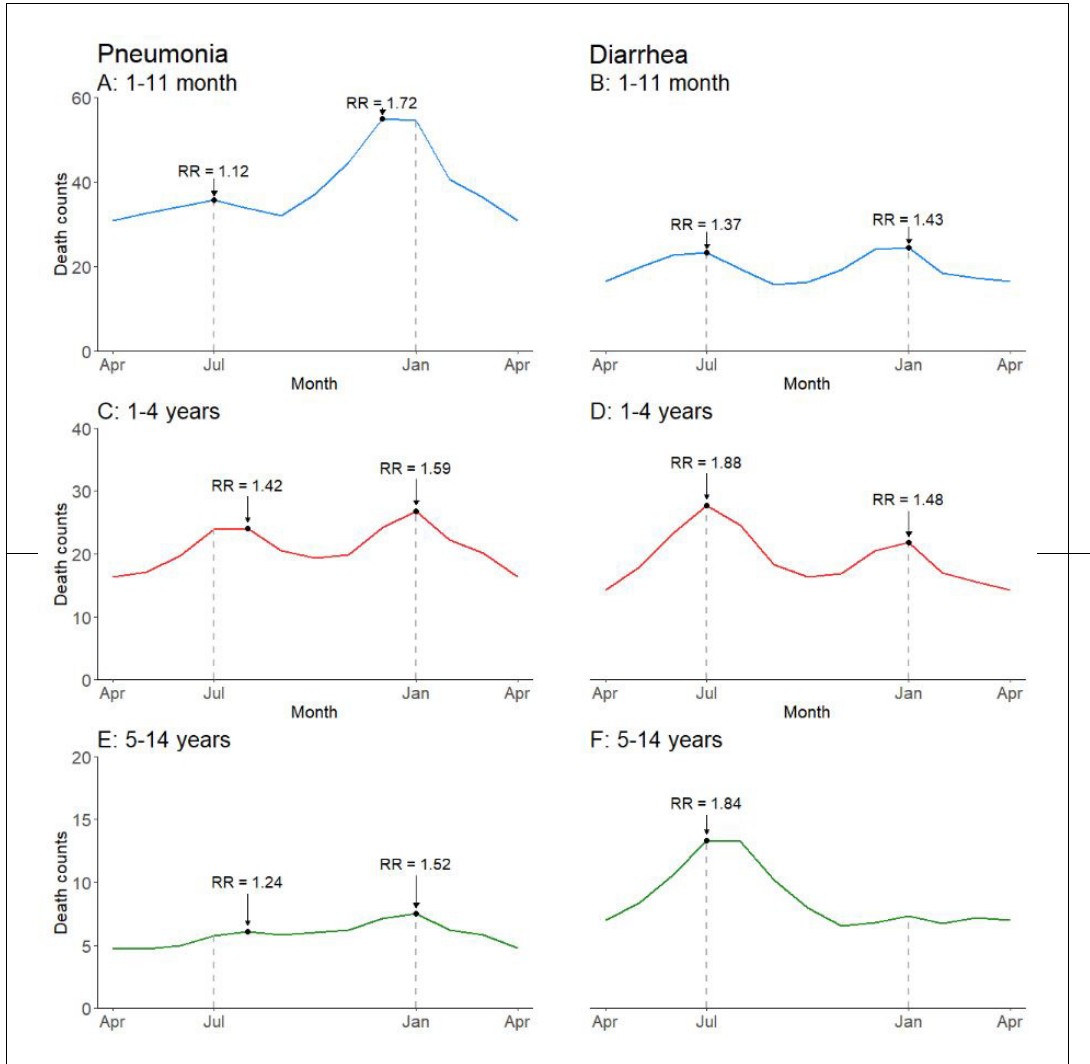

**Figure 4.** Average seasonal patterns of pneumonia (**A, C, E**) and diarrhea lacking fever and bloody stool (**B, D, F**) deaths by child age subset. Counts of diarrhea with fever and bloody stool were too small to model. Each horizontal axis represents an average yearly span from April to April. We determined seasonal patterns using monthly counts of death and modeled using Poisson regression. Rate ratios (RR) were calculated within each disease and were compared to annual minimum mortality in the month of April.

DOI: https://doi.org/10.7554/eLife.46202.013

The following source data and figure supplements are available for figure 4:

**Source data 1.** Monthly predicted values and rate ratios of the average annual pattern of pneumonia and diarrhea mortality, by child age subset.

DOI: https://doi.org/10.7554/eLife.46202.016

**Figure supplement 1.** Time series model for (**A**) pneumonia and (**B**) diarrhea, by age subset.

DOI: https://doi.org/10.7554/eLife.46202.014

**Figure supplement 1—source data 1.** Predicted values for pneumonia and diarrhea mortality time series models by child age subset; 2005–2013.

DOI: https://doi.org/10.7554/eLife.46202.015

autopsies (mostly of home, medically unattended deaths) are by necessity crude in terms of defining exact etiologies (*Aleksandrowicz et al., 2014*). Despite the inherent misclassification of verbal autopsies, we demonstrate that carefully-conducted, nationally-representative verbal autopsies can be paired effectively with clinical syndromes, seasonality and climate region to yield novel insights

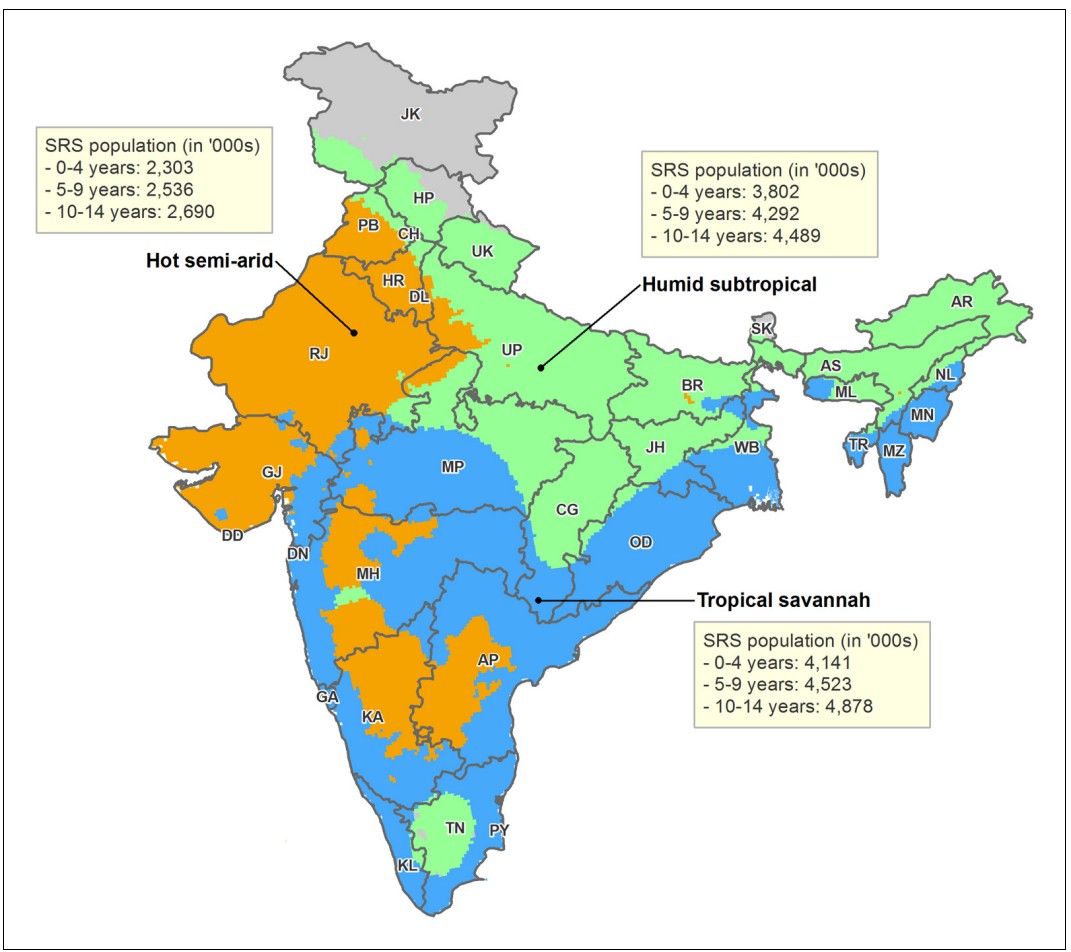

**Figure 5.** Map of India by state and Köppen-Geiger climate classification region. Original climate region data was extracted from *Kottek et al. (2006)*. The Köppen-Geiger map of India was created using ArcGIS 10.4. The three climate regions shown include hot semi-arid (Bsh; including hot desert or Bwh), humid subtropical (Cwa; including subtropical highland or Cwb), and tropical savannah (Aw; including tropical monsoon or Am). The hot desert, subtropical highland, and tropical monsoon regions are incorporated into similar, adjacent climate regions given insufficient sample size. Briefly, the hot semi-arid region exhibits large fluctuations in annual temperature, has little annual precipitation, and is geographically proximal to deserts; the tropical savannah region is characterized by distinct dry-wet seasonal variation; and the humid subtropical region is defined by hot, humid temperatures and consistent rainfall throughout the year. AN = Andaman and Nicobar Islands. AP = Andhra Pradesh. AR = Arunachal Pradesh. AS = Assam. BR = Bihar. CH = Chandigarh. CG = Chhattisgarh. DD = Daman and Diu. DN = Dadra and Nagar Haveli. DL = Delhi. GA = Goa. GJ = Gujarat. HP = Himachal Pradesh. HR = Haryana. JH = Jharkhand. JK = Jammu and Kashmir. KA = Karnataka. KL = Kerala. LD = Lakshadweep. MH = Maharashtra. ML = Meghalaya. MN = Manipur. MP = Madhya Pradesh. MZ = Mizoram. NL = Nagaland. OD = Odisha. PB = Punjab. PY = Puducherry. RJ = Rajasthan. SK = Sikkim. TN = Tamil Nadu. TR = Tripura. UP = Uttar Pradesh. UT = Uttarakhand. WB = West Bengal.

DOI: https://doi.org/10.7554/eLife.46202.017

into some of the etiologies of pneumonia and diarrhea, the two most common killers of children in the world.

The incidence of pneumonia mortality was 66% greater in January than the annual minimum in April. Infants aged 1–11 months contributed disproportionately to excess mortality in December and January, and viral etiologies such as RSV and influenza may contribute to this peak as they are known causes of respiratory infection in this age group during cooler months (*Agrawal et al., 2009*; *Tang and Loh, 2014*). Most RSV surveillance studies show the highest percent positivity between September and January (*Benet et al., 2017*; *Bharaj et al., 2009*; *Choudhary et al., 2013*),

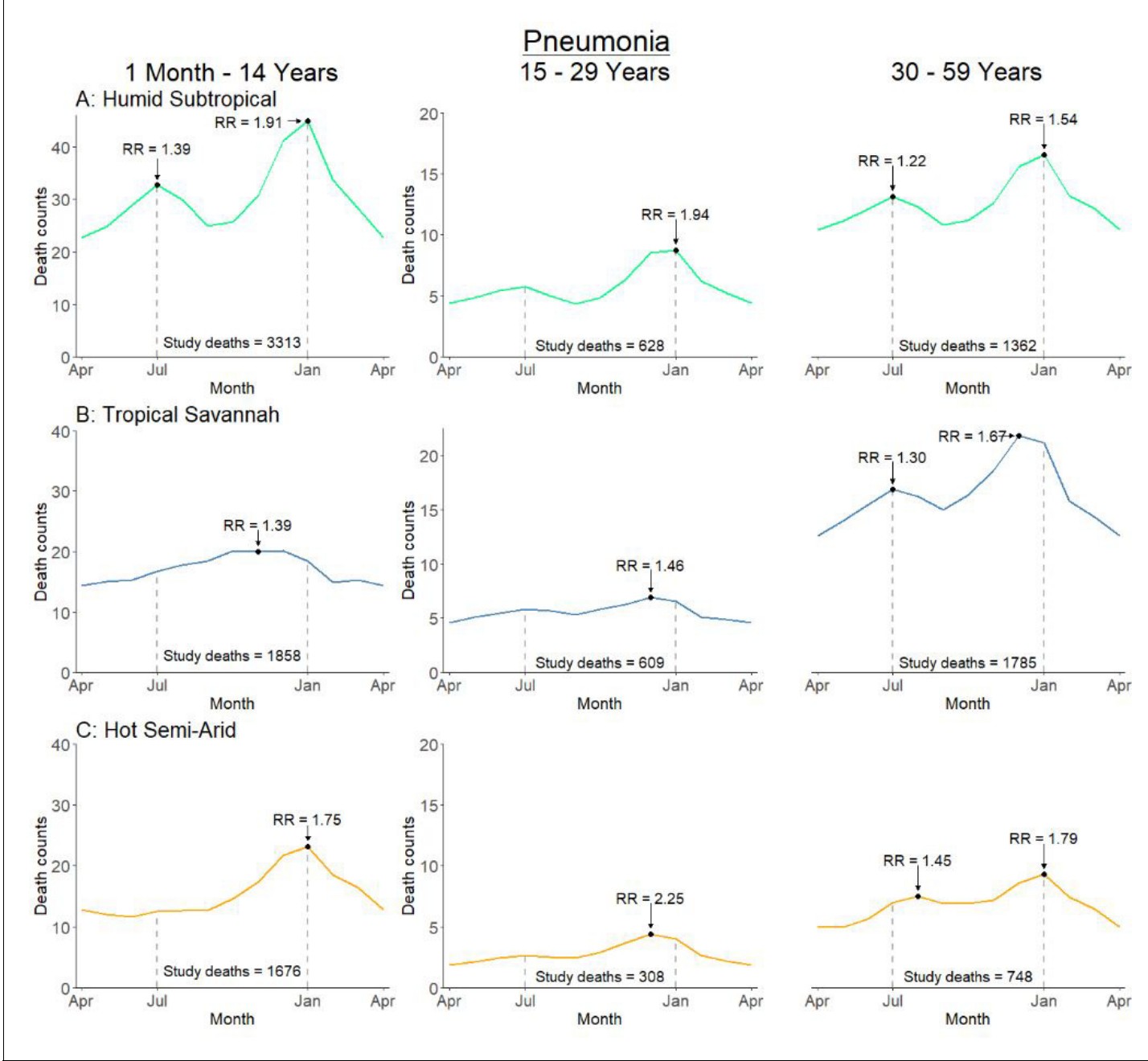

**Figure 6.** Average seasonal pattern of deaths from pneumonia by age group and Köppen-Geiger climate region. Each horizontal axis represents an average yearly span from April to April. We determined seasonal patterns using monthly counts of death and modeled using Poisson regression. Rate ratios (RR) were calculated within each disease and were compared to annual minimum mortality in the month of April. Given three regions also had a smaller sample size on which to model, they were bundled into regions with similar climatic characteristics (tropical monsoon into tropical savannah, hot desert into hot semi-arid, and subtropical highland into humid subtropical).

DOI: https://doi.org/10.7554/eLife.46202.018

The following source data and figure supplements are available for figure 6:

**Source data 1.** Monthly predicted values and rate ratios of the average annual pattern of pneumonia mortality, by age subset and climate region.
DOI: https://doi.org/10.7554/eLife.46202.021
**Figure supplement 1.** Time series model for (A) pneumonia and (B) diarrhea by Köppen-Geiger climate region.
DOI: https://doi.org/10.7554/eLife.46202.019
**Figure supplement 1—source data 1.** Predicted values for pneumonia and diarrhea mortality time series models by climate region; 2005–2013.
DOI: https://doi.org/10.7554/eLife.46202.020

suggesting that RSV may be the primary or indirect cause of much of the excess pneumonia mortality in children during these months. Viral respiratory infections may precede bacterial pneumonia in children, and direct association has been established between RSV and influenza and invasive pneumococcal disease (*Talbot et al., 2005*; *Weinberger et al., 2015*). Globally, the prevalence of pneumococcal pneumonia is also known to peak in cool, dry months, though very little data describing pneumococcal seasonality is available from India and this warrants further study, ideally through multi-center surveillance studies (*Numminen et al., 2015*). Notably, no licensed vaccine for RSV currently exists and only Novavax's RSV F maternal immunization has yet reached Phase three trials (*Novavax, Inc, 2019*; *PATH, 2018*). Hence, future introduction of RSV vaccines may be monitored indirectly by examining peak seasonal patterns by age and region.

Our climate region analysis demonstrates the July pneumonia peak evident nationally is mostly a result of a previously undocumented and unique peak in the humid subtropical region. We estimated that this peak was associated with 3500 excess deaths between April and September in 2015, though cannot define the etiology of this unique peak. However, the humid subtropical region is characterized by hot, humid summers, mild winters, and precipitation consistent throughout the year, conditions associated with viral etiologies such as influenza and RSV in tropical climates (*Kamigaki et al., 2016*; *Paynter, 2015*; *Tang and Loh, 2014*). Notably, adults aged 30–59 years exhibited excess mortality occurring around July, but these deaths were dispersed across all climate regions. The etiology of adult pneumonia deaths is more uncertain, in part as the agreement between dual independent physicians was lower than for children and because of the possible role of risk factors such as smoking and exposure to particulate matter (*Jha et al., 2008*; *Kim et al., 2017*).

The incidence of diarrhea mortality was 66% greater in July and 37% greater in January compared to the annual minimum. Previous evidence suggests that mid-year (July) diarrhea peaks are attributable to bacterial pathogens such as cholera and campylobacter species and parasitic pathogens such as cryptosporidium species whereas late-year (December-January) peaks are mostly attributable to rotavirus and other viral pathogens (*Ajjampur et al., 2010*; *Fletcher et al., 2013*; *Sebastian et al., 2015*). Diarrheal deaths with both fever and bloody stool may be clinically representative of enteroinvasive bacterial diarrhea, including microbiologic agents such as *Shigella spp.* and enteroinvasive *E. coli* (*Liu et al., 2016*).

Our age-stratified models suggest that rotavirus infection is the primary cause of diarrhea mortality during the December-January mortality peak. Though rotavirus infection circulates in India year-round, percent positivity among clinical cases is highest in cooler months between September and February (*Mehendale et al., 2016*). Rotavirus death is most common among children < 2 years but has a lower case-fatality rate at older ages, consistent with no observed December-January peak among children aged 5–14 years. Thus, we expect rotavirus deaths occurred disproportionately among the 58,600 all-cause diarrhea deaths that occurred in 2015 at ages 1–23 months. An earlier study using the MDS examined rotavirus mortality also at ages 1–59 months (*Morris et al., 2012*). The comparisons over time suggest about a 78% decline in rotavirus mortality (113,000 in 2005 versus 24,700 in 2013) versus a slightly smaller 67% decline in overall diarrheal mortality (334,000 versus 109,000). These declines occurred prior to the introduction of the Indian-made, low-cost oral rotavirus vaccine (RotaVac) in 2016. Hence, future diarrhea mortality may be expected to fall even faster. Future MDS data on age-specific and syndrome-specific (i.e. without fever and bloody stool) can help to assess these changes.

The MDS study enables comparison of mortality attributable to five microbiologic agents and two syndromes of pneumonia and diarrhea to modeled data. Compared to under-5 mortality estimates from the model-based Global Burden of Disease (GBD), we estimated 51% fewer pneumococcal pneumonia deaths and 66% fewer Hib pneumonia deaths in 2015 (*GBD 2015 LRI Collaborators, 2017*), though our estimates notably lack mortality from the first month of life. Though GBD estimates also utilize the vaccine probe approach used in this study, they also adjusted pneumococcal pneumonia estimates to a randomized controlled trial conducted in an elderly population from the Netherlands. This largely accounts for their etiologic fraction being twice as high as our estimate. For Hib pneumonia, GBD include Hib vaccine coverage as a covariate in their model rather than accounting for Hib vaccine coverage in a deterministic manner. This could lead to differences in the Hib pneumonia mortality estimates. Compared to the Maternal Child Epidemiology Estimation (MCEE) collaboration, we estimate 29% fewer pneumococcal pneumonia deaths and 40% fewer Hib

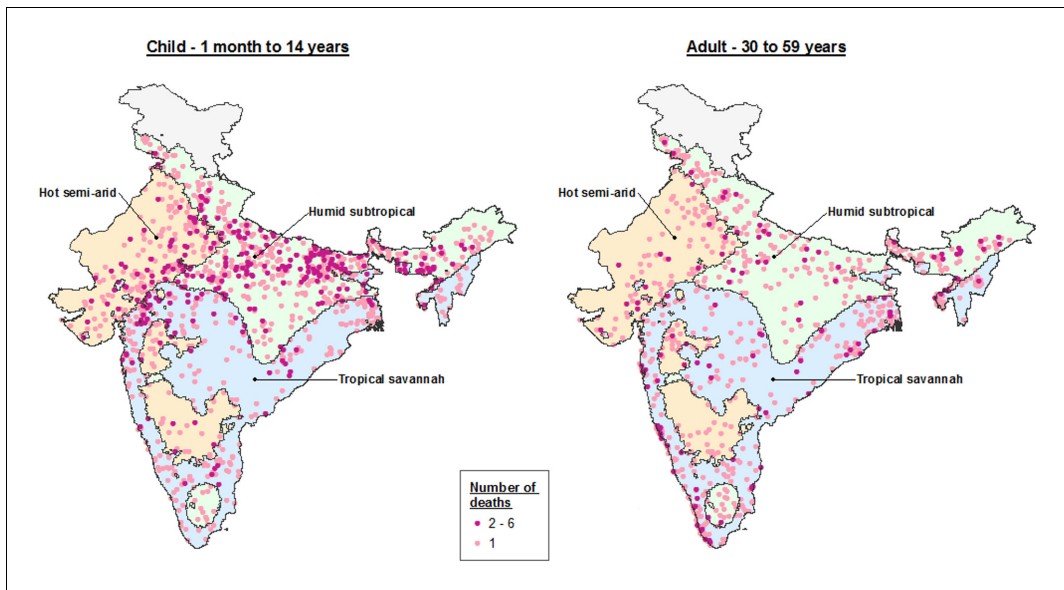

**Figure 7.** Dot map of child (1 month – 14 years) and adult (30–59 years) pneumonia deaths captured within the Million Death Study between June and August, by Köppen-Geiger climate region. The three climate regions shown include hot semi-arid (Bsh; including hot desert or Bwh), humid subtropical (Cwa; including subtropical highland or Cwb), and tropical savannah (Aw; including tropical monsoon or Am). The hot desert, subtropical highland, and tropical monsoon regions are incorporated into similar, adjacent climate regions given insufficient sample size to describe separately.

DOI: https://doi.org/10.7554/eLife.46202.022

pneumonia deaths in 2015 (*Wahl et al., 2018*). These differences are likely due to the higher estimates of all-cause pneumonia deaths by the MCEE collaboration.

By contrast, we estimated 146% more RSV pneumonia deaths and 425% more influenza deaths in 2015 than the GBD, as the modeled etiologic fractions in the GBD were calculated using a substantially lower proportion of test-positive cases than estimated in our meta-analyses (*GBD 2015 LRI Collaborators, 2017*). Our estimates of RSV and influenza are based on meta-analyses of directly tested specimens of hospitalized children in India. This has obvious limitations, but likely no more so than modeling results from the GBD (*Rigby et al., 2019*). The MDS study may be underestimating RSV mortality as we excluded neonates who are affected by this virus (*Caballero and Polack, 2018*; *Scheltema et al., 2017*). Our estimate of influenza mortality is also 50% lower than the recent nationally representative estimate of 27,825 deaths in a recent meta-analysis (*Ram Purakayastha et al., 2018*). However, those estimates also include neonates and use meta-analysis of studies outside India. For diarrhea, our estimates of rotavirus mortality are broadly consistent with GBD (*Troeger et al., 2017*) and our estimate for diarrhea cases with fever and bloody stool is similar to the GBD estimate of 4600 deaths from *Shigella spp.* (*Khalil et al., 2018*), suggesting that enteroinvasive bacterial pathogens are only a small minority of child diarrhea deaths in India (*Troeger et al., 2017*).

This research has several limitations. Firstly, our use of verbal autopsy may lead to uncertainties in cause of death coding. However, verbal autopsy remains the only viable way of monitoring causes of death in representative populations over time in India, and captures any changes since 2005 in oral rehydration treatment and improved access to care that may have reduced pneumonia and diarrhea mortality (*Fadel et al., 2017*). Moreover, our use of dual physician coding and adjudication minimizes potential misclassification bias and coding practices did not change over time. Notably, there was high agreement (>75%) for child pneumonia and diarrhea deaths while there was poor agreement for adult pneumonia deaths (38%). As such, greater misclassification may occur for adults due to the more disparate pathways of exposure and clinical disagreement resulting from potential comorbidities and further study of adult pneumonia is warranted. Secondly, our etiology-specific mortality calculations used several assumptions. Because in most cases verbal autopsy is unable to

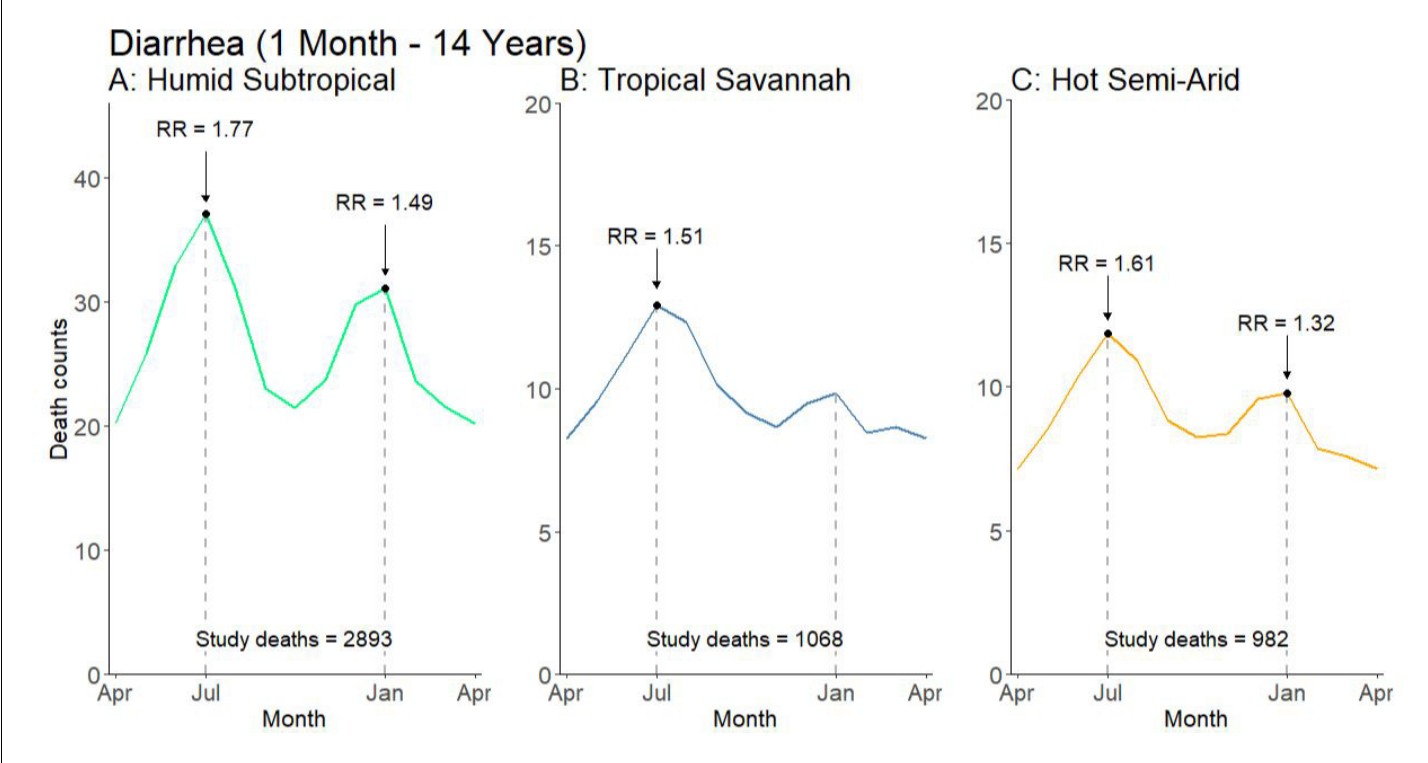

**Figure 8.** Average seasonal pattern of deaths from diarrhea lacking fever and bloody stool by Köppen-Geiger climate region among children at ages 1 month to 14 years. Counts of diarrhea with fever and bloody stool were too small to model. Each horizontal axis represents an average yearly span from April to April. We determined seasonal patterns using monthly counts of death and modeled using Poisson regression. Rate ratios (RR) were calculated within each disease and were compared to annual minimum mortality in the month of April. Given three regions also had a smaller sample size on which to model, they were bundled into regions with similar climatic characteristics (tropical monsoon into tropical savannah, hot desert into hot semi-arid, and subtropical highland into humid subtropical).

DOI: https://doi.org/10.7554/eLife.46202.023

The following source data is available for figure 8:

**Source data 1.** Monthly predicted values and rate ratios of the average annual pattern of child diarrhea mortality by climate region.
DOI: https://doi.org/10.7554/eLife.46202.024

assign microbiologic etiology, our regional meta-analyses relied on prevalence surveys among hospitalized children and children seeking care in emergency departments. This will naturally differ from prevalence among dead children, but we believe that given most of the non-fatal cases comprised severe disease, that differences should be small with deceased children (*Morris et al., 2012*). Additionally, we generated EFs for influenza and RSV using data collected between 2005–2013 while we used pneumonia mortality envelopes from 2015. It is therefore possible that mortality trends attributable to these conditions diverged from those of all-cause pneumonia during this time. The health centers included in our meta-analyses were also almost entirely located in urban areas and the etiologic fraction may differ in more remote areas. However, we believe that for most regions the aggregation of studies through meta-analysis accounts for these regional differences. We were also unable to disaggregate deaths due to pandemic influenza and seasonal influenza A and B. Thirdly, *Streptococcus pneumoniae* and Hib also contribute to bacterial sepsis and meningitis, and these conditions would also be reduced with wide coverage of the newly-available conjugate vaccines (*Wahl et al., 2018*). Fourth, we are unable to attribute declining mortality to either a lower incidence of disease or a lower case-fatality rate. Fifth, though climate dynamics were a heavy focus of this research, we could not assess important risk factors such as malnutrition and zinc deficiency. Finally, we found no evidence of shifting seasonal patterns over the study period of 2005–13, though future seasonal patterns may be affected by changing socioeconomic conditions, climatic changes, and global warming (*Burkart et al., 2011*; *Prüss-Ustün et al., 2017*).

**Table 3.** Estimated deaths due to microbiologic etiologies and other syndromes of pneumonia and diarrhea in 2015 among Indian children aged 1–59 months.

| Etiology | Source of estimation | Etiologic fraction (%) (uncertainty range) | Estimated deaths (uncertainty range) |
|---|---|---|---|
| *Streptococcus pneumoniae*[1] | Meta-analysis (clinical trials) | 38 (27–39) | 40,600 (28,800–42,400) |
| Respiratory syncytial virus[2] | Meta-analysis (prevalence surveys) | 19 (13–27) | 20,700 (13,600–29,000) |
| Influenza[2,3] | Meta-analysis (prevalence surveys) | 12 (7–19) | 12,600 (7,200–20,600) |
| *Haemophilus influenzae* type b[1] | Meta-analysis (clinical trials) | 7 (5–9) | 7200 (5,000–9,400) |
| Humid subtropical summer excess | Syndromic mortality estimate | 3 (no range) | 3500 (no range) |
| Pneumonia total[4] | | | 108,000 |
| Rotavirus[5] | Meta-analysis (prevalence surveys) | 30 (21–40) | 24,700 (17,200–32,800) |
| Diarrhea with fever and bloody stool | Syndromic mortality estimate | 8 (no range) | 6200 (no range) |
| Diarrhea total[4] | | | 82,000 |

[1]The proportion of pneumonia cases due to *Streptococcus pneumoniae* (pneumococcal) and *Haemophilus influenzae* type b (Hib) were determined using efficacy results from pneumococcal conjugate vaccine and Hib vaccine clinical trials. Proportions were then multiplied by all-cause pneumonia deaths among children aged 1–59 months. [2]The proportion of pneumonia cases due to RSV and influenza was determined through a literature search of laboratory testing and surveillance studies containing etiologic fractions. Etiologic fractions were then meta-analyzed by region, and then multiplied by region-specific all-cause pneumonia deaths. [3]Influenza includes influenza A (seasonal and pandemic) and influenza B. 4All-cause pneumonia and diarrhea deaths were previously reported in *Fadel et al. (2017)*. Etiologic fractions between pathogens are not additive and do not have a sum of 100. [5]The proportion of diarrhea cases due to rotavirus was determined through a literature search of laboratory testing and surveillance studies containing etiologic fractions. Etiologic fractions were then meta-analyzed by region, and then multiplied by region-specific all-cause diarrhea deaths.

DOI: https://doi.org/10.7554/eLife.46202.025

Notwithstanding these constraints, we believe our estimates in *Table 3* to be the best available estimates of the etiology-specific mortality burden in India. Our results emphasize the need for more robust surveillance systems such as the Expanded National Rotavirus Surveillance System (*Mehendale et al., 2016*; *INSPIRE investigators et al., 2017*). They also emphasize the underappreciated robustness of carefully conducted and standardized verbal autopsies done on nationwide, representative populations (*Jha, 2014*). Mortality and surveillance data combined with seasonality, clinical syndromes, and examined over space and time can provide novel insights into specific etiologies of pneumonia and diarrhea and enable appropriate introduction of vaccination and treatment strategies.

# Materials and methods

## Study design

In this study, we compiled mortality data for years 2005–2013 from the Million Death Study, a nationally representative survey conducted within India's Sample Registration System (SRS). The SRS is a continuous surveillance system established by the Registrar General of India (RGI) to provide reliable and routine data regarding vital status. The sampling design for the present study is a stratified random sample of over 1 million areas defined in the 2001 Indian Census, containing 150–300 homes in each area. In total, our study sample comprises 7597 areas (called sampling units), where the RGI administers surveys every 6 months to record vital events. For each death, trained surveyors interview close relatives of the deceased or other household members using a modified version of the 2012 WHO verbal autopsy questionnaire. The interview contains both a structured checklist of symptoms and a half-page open-ended narrative in the local language to ascertain the chronology of events which lead to death. Two of 400 trained and certified physicians independently assess verbal

**Table 4.** Age-specific mortality rates and etiologic fractions of microbiologic etiologies and other syndromes of pneumonia and diarrhea in 2015 among Indian children aged 1–59 months.

| Region | Pneumonia mortality rate (per 1000 live births)[1] | Etiologic fraction (%), (uncertainty range)[2] | | | | Diarrhea mortality rate (per 1000 live births)[1] | Etiologic fraction (%), (uncertainty range)[2] | |
| | | SPn[3] | RSV[4] | Influenza[4] | Hib[3] | | Rotavirus[4] | Diarrhea with fever and bloody stool[3] |
|---|---|---|---|---|---|---|---|---|
| Northeast | 9.50 | 40 (28–41) | 11 (8–15) | 6 (<1–17)[4] | 7 (5–9) | 6.96 | 41 (33–49) | 12 |
| Central | 5.36 | 38 (27–40) | 22 (15–30) | 16 (11–21) | 9 (6–12) | 3.80 | 22 (16–27) | 9 |
| East | 4.40 | 41 (29–43) | 12 (5–20) | 6 (<1–17) | 4 (3–6) | 4.38 | 34 (19–52) | 5 |
| North | 3.01 | 42 (30–44) | 25 (20–30) | 9 (3–18) | 3 (2–4) | 2.57 | 46 (37–56) | 8 |
| West | 1.77 | 40 (29–43) | 18 (16–21) | 9 (6–14) | 6 (4–8) | 1.16 | 33 (25–41) | 7 |
| South | 1.63 | 41 (29–43) | 35 (23–47) | 13 (10–17) | 5 (3–6) | 0.82 | 35 (30–41) | 4 |
| India | 4.17 | 38 (27–39) | 19 (13–27) | 12 (7–19) | 7 (5–9) | 3.19 | 30 (21–40) | 8 |
| EF Ratio (Highest: Lowest) | | 1.11 | 3.18 | 2.67 | 3.00 | | 2.09 | 3.00 |
| Pearson (vs. pneumonia) | | −0.36 | −0.66 | 0.20 | 0.46 | Pearson (vs. diarrhea) | 0.12 | 0.73 |

[1]1–59-month mortality rates were derived using estimates from *Fadel et al. (2017)*. [2]Etiologic fractions between pathogens are not additive and do not have a sum of 100. [3]Etiologic fractions of *Streptococcus pneumoniae* (SPn), *Haemophilus influenzae* type b (Hib), and diarrhea with fever and bloody were calculated by dividing the estimated number of deaths by pathogen and administrative region by the total number of pneumonia or diarrhea deaths by administrative region. [4]Etiologic fractions for respiratory syncytial virus (RSV), influenza, and rotavirus were determined with a meta-analysis of laboratory testing data by region. [5]Given that no studies were identified describing influenza positivity in the Northeast region, the etiologic fraction for the East region (closest geographically) is substituted instead.

DOI: https://doi.org/10.7554/eLife.46202.026

The following source data is available for Table 4:

**Source data 1.** Regional meta-analysis of etiologic fractions for respiratory syncytial virus.

Studies reporting laboratory testing data were identified through a literature search of Ovid MEDLINE, Scopus, and Google Scholar. All studies described hospitalized children or children seeking care in emergency departments and reported data from 2005 onwards. Studies were meta-analyzed by administrative region using Stata's metaprop package and visualized in RStudio. We weighted each study using the denominator of total number of laboratory tests in the respective study.

DOI: https://doi.org/10.7554/eLife.46202.027

**Source data 2.** Regional meta-analysis of etiologic fractions for influenza.

Studies reporting laboratory testing data were identified through a literature search of Ovid MEDLINE, Scopus, and Google Scholar. All studies described hospitalized children or children seeking care in emergency departments and reported data from 2010 onwards. All studies included describe pandemic and seasonal influenza A and influenza B. Studies were meta-analyzed by administrative region using Stata's metaprop package and visualized in RStudio. We weighted each study using the denominator of total number of laboratory tests in the respective study.

DOI: https://doi.org/10.7554/eLife.46202.028

**Source data 3.** Regional meta-analysis of etiologic fractions for rotavirus.

Studies reporting laboratory testing data were identified through a literature search of Ovid MEDLINE, Scopus, and Google Scholar. All studies described hospitalized children or children seeking care in emergency departments and reported data from 2010 onwards. Studies were meta-analyzed by administrative region using Stata's metaprop package and visualized in RStudio. We weighted each study using the denominator of total number of laboratory tests in the respective study.

DOI: https://doi.org/10.7554/eLife.46202.029

autopsy reports and assign a standardized cause of death according to the International Classification of Diseases and Related Health Problems, 10th Revision (ICD-10) (*World Health Organization, 2010*). Initial disagreement in cause of death coding is then reconciled anonymously, and a third, more senior physician adjudicates any persisting disagreements to determine the final cause of death. Greater detail regarding MDS study design and protocol have been published elsewhere (*Aleksandrowicz et al., 2014*; *Jha et al., 2006*).

We identified pneumonia deaths from the MDS using ICD-10 codes A37, H65-H68, H70, H71, J00-J22, J32, J36, J85, J86, P23, and U04, and diarrhea deaths as codes A00 and A02-A09. We excluded 272 deaths from our pneumonia case definition and 132 child deaths from our diarrhea case definition which had a reported history of measles by the VA respondent, in an effort to

minimize misclassification of deaths which may be secondary to a measles infection (*Wong et al., 2019*). We excluded an additional 557 child deaths due to typhoid or paratyphoid fever (ICD-10 code A01) from our diarrhea case definition, as these cases do not share a common symptom profile with other infectious causes of diarrheal disease (*Morris et al., 2011*). We disaggregated child diarrhea into two sub-groups using a symptom profile defined by cases with both reported fever and bloody stool (*Liu et al., 2016*). We define cases with these symptoms as enteroinvasive bacterial diarrhea and those without as all other diarrhea. Adult pneumonia deaths (15–69 years) were identified using the same ICD-10 codes and did not exclude measles. Finally, we excluded neonatal deaths from this study as the neonatal VA administered may be less accurate in identifying pneumonia and diarrhea deaths, infection pathways may differ between neonatal and post-neonatal etiologies, and because standard childhood antigens (e.g. rotavirus, Hib conjugate, pneumococcal conjugate) are all first administered after the neonatal period.

## Mortality trends

We proportionally adjusted total live births and deaths to reflect the 2015 population estimates from the United Nations Population Division and the Inter-agency Group for Child Mortality Estimation. We stratified data by sex, urban/rural residence (areas defined by the RGI), and residence in an Empowerment Action Group-Assam state (the nine poorest states in India compared to all other states and union territories). All proportions were weighted by SRS sampling probability to adjust for variation in sampling differences between rural and urban areas. We calculated the number of deaths and mortality rates using a three-year moving average of the weighted proportion of deaths, using a denominator of live births for ages 1–59 months and population for ages 5–14 years. We then calculated absolute mortality rate reductions between 2005 and 2013 using a log percentage change method (*You et al., 2015*).

## Time series analysis to document seasonality

We assessed seasonality of pneumonia and diarrhea using a time series analysis (*Bhaskaran et al., 2013*). Using monthly counts of death, we applied generalized linear models with a Poisson distribution, two sine-cosine pairs per year and an offset of the number of days per month. We computed rate ratios for each model comparing peak mortality to the number of deaths in April, the month in which mortality was lowest for both conditions. We also applied generalized additive models with a cubic spline smoothing component to account for non-linear trends in the data. We added a national comparison of typhoid and paratyphoid fever deaths (ICD-10 code A02). We also completed age-stratified models to identify differences in the seasonal pattern for children at ages 1–11 months, 1–4 years, and 5–14 years and supplemented these with comparative analyses for adults aged 15–29 years, 30–59 years, and 60–69 years for pneumonia only.

We assessed geographic differences in seasonality by climate region, defined using the Köppen-Geiger climate classification system (*Kottek et al., 2006*). This system assigns climate zones based on annual patterns of temperature and rainfall (*Kottek et al., 2006*). We analyzed six climate regions using the above described modeling techniques, including the tropical monsoon (class Am), tropical savannah (class Aw), hot semi-arid (class Bsh), hot desert (class Bwh), humid subtropical (class Cwa), and subtropical highland (class Cwb) regions. These six regions represented 96% of all study pneumonia deaths and 95% of all study diarrhea deaths. We combined regions with an insufficient sample size for modeling into similar, more populous regions (tropical monsoon into tropical savannah, hot desert into hot semi-arid, and subtropical highland into humid subtropical). Briefly, the tropical savannah region is characterized by distinct dry-wet seasonal variation; the hot semi-arid region exhibits large fluctuations in annual temperature, has little annual precipitation, and is geographically proximal to deserts; and the humid subtropical region is defined by hot, humid temperatures and consistent rainfall throughout the year. Again, we conducted supplemental comparative analyses for adults aged 15–29 years and 30–59 years for pneumonia only. Adults aged 60–69 years were not included due to insufficient sample sizes.

## Estimation of etiology-specific mortality

We estimated the number of deaths among children aged 1–59 months for pneumonia attributable to Hib, influenza, pneumococcal, and RSV infection and diarrhea due to rotavirus infection using an

attributable fraction framework. For influenza, RSV, and rotavirus infection, we generated etiologic fractions using a meta-analysis of data from published studies describing the proportion of specimens testing positive for these pathogens in Indian settings. For Hib and pneumococcal infection, we used the probe approach and Hib vaccine and PCV clinical trial data from around the world to calculate Hib and pneumococcal pneumonia mortality etiologic fractions. We then applied etiologic fractions to the number of all-cause pneumonia and diarrhea deaths among children aged 1–59 months as presented by *Fadel et al. (2017)*. Greater detail regarding these calculations is provided in supplemental text.

Analyses were performed using Stata version 15.1 and R version 3.4.1.

## Acknowledgements

The authors would like Patrick Brown and Wilson Suraweera for providing statistical and data oversight, and to Umesh Parashar for his comments on the working draft manuscript.

## Additional information

### Competing interests

Prabhat Jha: Reviewing editor, *eLife*. The other authors declare that no competing interests exist.

### Funding

| Funder | Grant reference number | Author |
| --- | --- | --- |
| Canadian Institutes of Health Research | FDN154277 | Prabhat Jha |
| Bill and Melinda Gates Foundation | | Prabhat Jha |
| National Institutes of Health | R01TW05991-01 | Prabhat Jha |

The funders had no role in study design, data collection, analysis or interpretation, preparation of the manuscript or the decision to submit the work for publication.

### Author contributions

Daniel S Farrar, Resources, Data curation, Formal analysis, Validation, Investigation, Visualization, Methodology, Writing—original draft, Project administration, Writing—review and editing; Shally Awasthi, Conceptualization, Resources, Investigation, Project administration, Writing—review and editing; Shaza A Fadel, Conceptualization, Resources, Data curation, Supervision, Investigation, Methodology, Project administration, Writing—review and editing; Rajesh Kumar, Anju Sinha, Resources, Project administration, Writing—review and editing; Sze Hang Fu, Resources, Data curation, Visualization, Methodology, Project administration, Writing—review and editing; Brian Wahl, Resources, Formal analysis, Project administration, Writing—review and editing; Shaun K Morris, Conceptualization, Resources, Methodology, Project administration, Writing—review and editing; Prabhat Jha, Conceptualization, Resources, Supervision, Funding acquisition, Validation, Investigation, Writing—original draft, Project administration, Writing—review and editing

### Author ORCIDs

Daniel S Farrar https://orcid.org/0000-0002-7823-1912
Shaza A Fadel http://orcid.org/0000-0002-2336-6254
Sze Hang Fu http://orcid.org/0000-0003-4890-9339
Prabhat Jha https://orcid.org/0000-0001-7067-8341

### Decision letter and Author response

Decision letter https://doi.org/10.7554/eLife.46202.034
Author response https://doi.org/10.7554/eLife.46202.035

# Additional files

## Supplementary files

• Source code 1. R code for pneumonia and diarrhea seasonality analyses.
DOI: https://doi.org/10.7554/eLife.46202.030

• Supplementary file 1. References for regional meta-analysis.
DOI:

• Transparent reporting form
DOI: https://doi.org/10.7554/eLife.46202.031

## Data availability

Data from the Million Death Study cannot be redistributed outside of the Centre for Global Health Research due to legal agreements with the Registrar General of India. Access to MDS data can be granted via data transfer agreements, upon request to the Office of the Registrar General, RK Puram, New Delhi, India (rgoffice.rgi@nic.in). The public census reports can be found at http://www.censusindia.gov.in/vital_statistics/SRS_Statistical_Report.html. Source data files have been provided for Figure 3, Figure 3—figure supplement 1, Figure 3—figure supplement 2, Figure 4, Figure 4—figure supplement 1, Figure 6, Figure 6—figure supplement 1, and Figure 8. Meta-analyses include only previously published data, and all data sources have been listed in supplemental reference lists within the article file.

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

## Appendix 1

DOI: https://doi.org/10.7554/eLife.46202.032

## Calculation of etiology-specific Mortality

We estimated microbiologic etiology-specific mortality of pneumonia due to Hib, influenza, pneumococcal, and RSV infection and diarrhea due to rotavirus infection using an attributable fraction framework. Briefly, we generated etiologic fractions at the administrative region level and multiplied these values by the state-specific pneumonia and diarrhea mortality in 2015 as presented by *Fadel et al. (2017)*.

We generated etiologic fractions for influenza, RSV, and rotavirus as follows. We reviewed Ovid MEDLINE, Scopus, and Google Scholar for studies describing the proportion of directly tested specimens positive for these microbiologic etiologies. As laboratory testing is rare for deceased children, we included only studies of hospitalized children or children seeking care in emergency departments. Therefore, we assumed the severity of cases and EFs were similar between these children and the deceased children identified in the MDS, as previously done (*Morris et al., 2012*). Additionally, we only selected studies which described populations < 5 years and conducted over a period of at least 12 months to account for seasonality. We did not limit our searches to specific case definitions or diagnostic methods. However, most included studies conducted sampling for RSV and influenza using nasopharyngeal swabs from children exhibiting severe acute respiratory infection, influenza-like illness, or pneumonia. Rotavirus sampling was largely conducted using stool samples from children exhibiting acute gastroenteritis. For influenza, we selected studies describing testing for any of influenza A (both pandemic and seasonal) or B. Given a limited number of available studies, we selected studies describing influenza and RSV data from 2005-onwards and rotavirus data from 2010-onwards. In cases where multiple studies included tests from the same hospital and timeframe, we selected the study with the most recent data. Next, we extracted the total number of positive samples and total number of samples tested to generate the proportion positive for each study. We then meta-analysed these proportions by India's six administrative regions, weighting the proportions by the total number of tested samples in the study. We did not meta-analyse studies by climate region as hospital catchment areas may have covered multiple regions, and subnational mortality estimates at the climate region level were not available. We conducted these meta-analyses using Stata's *metaprop* package and visualized forest plots with R.

To estimate state-level mortality attributable to these three microbiologic etiologies, we multiplied the region-specific etiologic fractions calculated above by the 2015 state-specific mortality of all-cause pneumonia and all-cause diarrhea presented in *Fadel et al. (2017)*. We then summed these state-specific estimates together to generate national mortality attributable to influenza, RSV, and rotavirus. Finally, we computed the national etiologic fraction by dividing the estimated number of microbiologic etiology-specific deaths by the total number of all-cause pneumonia or diarrhea deaths in *Fadel et al. (2017)*.

There remain several challenges associated with using observational studies to determine the fraction of pneumonia cases due to pneumococcus and Hib (*Levine et al., 2009*; *Moïsi et al., 2009*). We instead estimated the state-level burden of pneumonia mortality attributable to these pathogens using methods that have been reported previously for India (*Wahl et al., 2019*). Briefly, we identified efficacy data from pneumococcal conjugate vaccine (PCV) and Hib conjugate vaccine clinical trials conducted in diverse epidemiological settings around the world. Because no PCV or Hib vaccine clinical trials measured efficacy against all-pneumonia deaths, we used efficacy against chest radiograph confirmed pneumonia as a proxy for the efficacy against pneumonia deaths. To estimate the proportion of total pneumonia deaths due to each pathogen using the probe approach, we adjusted for: 1) incomplete vaccine efficacy using the efficacy against confirmed serotype-specific invasive pneumococcal disease from each trial, 2) the proportion of serotypes that cause disease included in the vaccine for PCV from the control group in the study, and 3) for pneumococcus,

the proportion of pneumonia deaths caused by Hib since all PCV trials were conducted in the presence of Hib vaccine. A meta-analysis summary estimate was calculated using Stata's *metaprop* package. These proportions were applied to all-cause pneumonia deaths in each region in 2015 as described above. We did not take into account the seasonality associated with pneumococcal or Hib pneumonia deaths. We then accounted for Hib vaccine use in states where the vaccine was administered using coverage of diphtheria-tetanus-pertussis as a proxy, as these vaccines are delivered in combination in India. PCV was not routinely used in any states in 2015 and private sector use was very low, so no *post hoc* adjustment for its use was conducted. With these state-level mortality estimates, we summed the national mortality burden to pneumococcal pneumonia and Hib and generated the national etiologic fraction as above.

Finally, we also present etiologic fractions of two syndromes including the previously described diarrhea cases with fever and bloody stool. EFs at the national and administrative region level were computed by dividing the number of diarrhea deaths among children aged 1–59 months with reported fever and bloody stool by the total number of 1–59 month diarrhea deaths. Based on the results of our seasonality models, we also estimated the etiologic fraction of 1–59 month pneumonia cases occurring during the humid subtropical region's mid-year peak. We assumed a uniform change in rates of death between April and September (i.e. the annual minima) and computed the expected number of deaths per month using simple linear regression among ages 1–59 months only. Next, we determined the difference between the number of actual and expected cases between April and September and summed these values to generate the annual excess attributable to this peak. We then generated the proportion of national deaths attributable to this peak and divided this value by the total number of all-cause pneumonia deaths reported in *Fadel et al. (2017)* to generate the etiologic fraction. Notably, the above two estimates lack uncertainty bounds.

