## [Decision Letter]

Thank you for submitting your article "Seasonal variation and etiologic inferences of childhood pneumonia and diarrhea mortality in India" for consideration by *eLife*. Your article has been reviewed by Neil Ferguson as the Senior Editor, a Reviewing Editor, and two reviewers. The reviewers have opted to remain anonymous.

The reviewers have discussed the reviews with one another and the Reviewing Editor has drafted this decision to help you prepare a revised submission.

Summary:

This is a very well written paper that provides very useful breakdowns of causes of death by syndrome. It captures etiological faction, state level variations and climatic heterogeneity and its relationship to pneumonia and diarrhea deaths data from verbal autopsies.

Essential revisions:

1) However, difficulties arise when pathogen-specific estimates are stated. There is insufficient information provided that would enable the reader to see how, for instance, such a precise number as 40,600 deaths were attributed to *S. pneumoniae*. In the methods, you refer to a paper that provides estimates based on vaccine probe calculations (like in Table 3), but it is not transparent how these were applied, considering seasonality and environmental classifications, to the proportion of pneumonia deaths. A table or figure (or simply paragraphs in the Materials and methods section) this needs to be shown clearly for each pathogen assessed.

2) When we get into more granular discussions like in subsection “Estimation of Etiology-specific Mortality”, it is really unclear what the process and bases were to estimate the excess 3500 deaths compared to a counterfactual (which needs to be explained better). Likewise, precisely what data and processes were used to come up with 24,700 deaths due to rotaviruses. You should aim to differ from the "black box feel" to what often is published in global estimates – there is a need and opportunity for the reader to clearer understand how the calculations were derived. This will go a long way towards such estimates to be valued and ultimately used.

3) Importantly, the limitations section later in the manuscript does not go through the substantial limitations associated with the assumptions used to make these calculations. And uncertainty ranges around all of these estimates need to be at least alluded to.

4) This becomes particularly challenging in subsection “Estimation of Etiology-specific Mortality” (and Table 4) when etiologic fractions were stated to vary by region. Again, the considerations and calculations around these estimates need to be carefully shown and validated for the findings to be credible to many reviewers and knowledgeable readers. There is insufficient description of how this was done in the methods (subsection “Estimation of Etiology-specific Mortality”). While at it, not clear how calculations were done for statement in the third paragraph of subsection “Estimation of Etiology-specific Mortality”.

5) We question the statement in the first paragraph of the Discussion section that despite inherent misclassification of VA (a big issue which the authors only allude to here) that they can be paired with other data to yield novel (and valid) insights.

6) The second paragraph of the Discussion section is unsubstantiated. Pneumococcal disease is also more common in winter months, and seasonal distribution of other bacterial pathogens can also mimic. Thus, the musing about importance of viral infections and their preceding of bacterial infections are interesting but risky to relate to the findings here. Likewise, the statement “Hence, future introduction of RSV vaccines can be monitored indirectly by examining peak seasonal patterns by age and region” should really be qualified heavily with the need for corresponding pathogen detection – very risky in either direction (success or failure) to make such assumptions about whether a vaccine is effective without parallel surveillance to validate circulating seasonal viruses and other pathogens.

7) The fifth paragraph of the Discussion section raises concerns that this conclusion was self-fulfilling based on assumptions made. Concern is compounded by observation of such a dramatic fall in rotavirus-attributed disease over time before vaccine was introduced. That suggests a potential flaw in assumptions used, unless an explanation for such a drop could be provided – i.e. dramatically improved access to health care and to ORS? Again, the limitations need to include the major limitations associated with the assumptions first. In addition, the fifth assumption (Discussion section) should be moved higher. VA reliance should of course be listed as a major limitation. More humility in this limitations section is really needed to be able to make the claim stated in the last paragraph of the Discussion section.

8) One of the key findings of the research is January and July peak on pneumonia and diarrhea deaths. In light of the SRS survey design and methodology, especially six months recording of event and interview through WHO verbal autopsy tool, it is important to mention in detail in the methods section, the months in which interviews were conducted and what proportion of data is collected by what month and what methods were adopted to address recall bias in the data collection. Peaks in month of January and July for both disease under consideration requires elaborate explanation on methods to verify outcome of interest (pneumonia and diarrhea death) and measures to address recall bias.

9) An explanation is required on some of the reasons on why there is significant reduction (from 77.4% in Table1 to 37.9% in Table 2) in agreement between two the physicians for initial assignment for pneumonia. This has implications on further analysis and reporting that have been conducted for adult age subset, for example average seasonal patterns of pneumonia deaths in Figure 3. An explanation on how this disagreement could have biased the results and how it was resolved would improve the manuscript.

---

## [Author Response]

Essential revisions:1) However, difficulties arise when pathogen-specific estimates are stated. There is insufficient information provided that would enable the reader to see how, for instance, such a precise number as 40,600 deaths were attributed to S. pneumoniae. In the methods, you refer to a paper that provides estimates based on vaccine probe calculations (like in Table 3), but it is not transparent how these were applied, considering seasonality and environmental classifications, to the proportion of pneumonia deaths. A table or figure (or simply paragraphs in the Materials and methods section) this needs to be shown clearly for each pathogen assessed.

Thank you. We note that this comment and others (i.e. essential revisions #2 and #4) refer to insufficient description of our etiology-specific mortality calculations. We agree with the reviewer that it is important to transparently present the methods used to estimate pathogen-specific deaths.

We have added in the main text a brief description of the framework used to calculate etiology-specific mortality, as below. In addition, to address the reviewer’s comments we have composed an appendix which describes these calculations in substantially more detail. Notably, we specified here that etiologic fractions were applied to pneumonia and diarrhea deaths for all of 2015; that is, we did not take into account the seasonality of pathogen-specific deaths. Rather, we assume that the studies that inform the etiologic fraction estimates capture the diversity of seasonality for these pathogens and therefore it would be appropriate to apply these fractions to the 2015 deaths.

“We estimated the number of deaths among children aged 1–59-months for pneumonia attributable to Hib, influenza, pneumococcal, and RSV infection and diarrhea due to rotavirus infection using an attributable fraction framework. […] Greater detail regarding these calculations is provided in supplemental text.”

2) When we get into more granular discussions like in subsection “Estimation of Etiology-specific Mortality”, it is really unclear what the process and bases were to estimate the excess 3500 deaths compared to a counterfactual (which needs to be explained better). Likewise, precisely what data and processes were used to come up with 24,700 deaths due to rotaviruses. You should aim to differ from the "black box feel" to what often is published in global estimates – there is a need and opportunity for the reader to clearer understand how the calculations were derived. This will go a long way towards such estimates to be valued and ultimately used.

As noted above, we have revised the description of these calculations and greater details regarding these methods is now included in our supplemental text. The following sections specifically addresses the methods used to determine the 3500 excess deaths.

Results section: “We also estimated an excess of 3,500 deaths at ages 1–59 months from pneumonia in the humid subtropical region between April and September, compared to the expected number of deaths in the absence of mortality changes between these two annual lows.”

Appendix: “Based on the results of our seasonality models, we also estimated the etiologic fraction of 1–59-month pneumonia cases occurring during the humid subtropical region’s mid-year peak. […] Notably, the above two estimates lack uncertainty bounds.”

3) Importantly, the limitations section later in the manuscript does not go through the substantial limitations associated with the assumptions used to make these calculations. And uncertainty ranges around all of these estimates need to be at least alluded to.

Where possible, uncertainty bounds have been added for both etiologic fractions and etiology-specific mortality estimates throughout the manuscript.

We have also revised the discussion of our limitations section, describing more of the assumptions used within our etiology-specific mortality calculations:

Discussion section: “Secondly, our etiology-specific mortality calculations used several assumptions. […] We were also unable to disaggregate deaths due to pandemic influenza and seasonal influenza A and B.”

4) This becomes particularly challenging in subsection “Estimation of Etiology-specific Mortality” (and Table 4) when etiologic fractions were stated to vary by region. Again, the considerations and calculations around these estimates need to be carefully shown and validated for the findings to be credible to many reviewers and knowledgeable readers. There is insufficient description of how this was done in the methods (subsection “Estimation of Etiology-specific Mortality”). While at it, not clear how calculations were done for statement in the third paragraph of subsection “Estimation of Etiology-specific Mortality”.

Thank you. Uncertainty bounds have been added to Table 3 and Table 4. Regarding the insufficient description of calculations, this has been addressed in the revised supplemental text as noted above for essential revision #1.

5) We question the statement in the first paragraph of the Discussion section that despite inherent misclassification of VA (a big issue which the authors only allude to here) that they can be paired with other data to yield novel (and valid) insights.

We have further described the potential biases associated with our use of verbal autopsy below. However, we believe the methods and results presented in this paper supplement the existing model-based child mortality literature, such as the Global Burden of Disease. Though we acknowledge verbal autopsy does not identify microbiologic etiology, we believe the nationally representative, directly captured death data allows insights to the national and subnational patterns of seasonality and speculation of microbiologic etiology in greater scrutiny than do model-based and other current studies.

Discussion section “Firstly, our use of verbal autopsy may lead to uncertainties in cause of death coding. […] As such, greater misclassification may occur for adults due to the more disparate pathways of exposure and clinical disagreement resulting from potential comorbidities and further study of adult pneumonia is warranted.”

6) The second paragraph of the Discussion section is unsubstantiated. Pneumococcal disease is also more common in winter months, and seasonal distribution of other bacterial pathogens can also mimic. Thus, the musing about importance of viral infections and their preceding of bacterial infections are interesting but risky to relate to the findings here. Likewise, the statement “Hence, future introduction of RSV vaccines can be 375 monitored indirectly by examining peak seasonal patterns by age and region” should really be qualified heavily with the need for corresponding pathogen detection – very risky in either direction (success or failure) to make such assumptions about whether a vaccine is effective without parallel surveillance to validate circulating seasonal viruses and other pathogens.

We thank the reviewer for this comment, which generated significant discussion among co-authors. We agree that pneumococcal disease is more common in winter months. However, the majority of studies describing pneumococcal seasonality originate from high income countries (i.e. United States), and these regions may not be representative of pneumococcal transmission in India due to differences in both wealth and climate effects. In a US study, Weinberger et al. (2014) also describe differential seasonal patterns of non-pneumonia pneumococcus and pneumococcal pneumonia, so it may be risky to draw conclusions regarding seasonality of pneumococcal pneumonia from studies describing all invasive pneumococcal disease. Very little data regarding the seasonality of pneumococcal pneumonia comes from India, though Awasthi et al. (1997) did describe a peak between December–February in Uttar Pradesh. This study also noted that *Streptococcus pneumoniae* was the predominant isolate in children >1 year, but not children <1 year.

Conversely, frequent and strong evidence regarding the seasonality of viral respiratory infections (i.e. RSV and influenza) is available in Indian studies. These studies, in tandem with the well-established link between viral and bacterial pneumonia, led us to our discussion regarding viral infections being the likely cause of December-January peaks in children <1 year. We absolutely agree that the seasonality of pneumococcal pneumonia in India requires significant further scrutiny. Considering all of these data, we have adjusted the Discussion section as follow:

“Infants aged 1–11 months contributed disproportionately to excess mortality in December and January, and viral etiologies such as RSV and influenza may contribute to this peak as they are known causes of respiratory infection in this age group during cooler months (Agrawal et al., 2009; Tang and Loh, 2014).”

“Globally, the prevalence of pneumococcal pneumonia is also known to peak in cool, dry months, though very little data describing pneumococcal seasonality is available from India and this warrants further study, ideally through multi-center surveillance studies.”

7) The fifth paragraph of the Discussion section raises concerns that this conclusion was self-fulfilling based on assumptions made. Concern is compounded by observation of such a dramatic fall in rotavirus-attributed disease over time before vaccine was introduced. That suggests a potential flaw in assumptions used, unless an explanation for such a drop could be provided – i.e. dramatically improved access to health care and to ORS? Again, the limitations need to include the major limitations associated with the assumptions first. In addition, the fifth assumption (Discussion section) should be moved higher. VA reliance should of course be listed as a major limitation. More humility in this limitations section is really needed to be able to make the claim stated in the last paragraph of the Discussion section.

As previously described, we have revised our Discussion section to describe the limitations of both verbal autopsy and our etiology-specific calculations in greater detail as follows:

“Firstly, our use of verbal autopsy may lead to uncertainties in cause of death coding. However, verbal autopsy remains the only viable way of monitoring causes of death in representative populations over time in India, and captures any changes since 2005 in oral rehydration treatment and improved access to care that may have reduced pneumonia and diarrhea mortality (Fadel et al., 2017). Moreover, our use of dual physician coding and adjudication minimizes potential misclassification bias and coding practices did change over time.”

“Secondly, our etiology-specific mortality calculations used several assumptions. […] However, we believe that for most regions the aggregation of studies through meta-analysis accounts for these regional differences.”

8) One of the key findings of the research is January and July peak on pneumonia and diarrhea deaths. In light of the SRS survey design and methodology, especially six months recording of event and interview through WHO verbal autopsy tool, it is important to mention in detail in the methods section, the months in which interviews were conducted and what proportion of data is collected by what month and what methods were adopted to address recall bias in the data collection. Peaks in month of January and July for both disease under consideration requires elaborate explanation on methods to verify outcome of interest (pneumonia and diarrhea death) and measures to address recall bias.

We acknowledge there may be a small bias to detecting an increased number of deaths in January and July, as the SRS survey design occurs on a biannual basis and is initiated in January and July. In this case, deaths from much earlier in the biannual period may not have been captured at a rate disproportionate to those occurring more recently to VA fieldwork dates. However, we strongly believe our results did not exhibit this bias. Firstly, though pneumonia and diarrhea exhibit marked seasonality in the MDS dataset, many other conditions do not (i.e. non-communicable diseases). Secondly, our results showed differences in seasonal variation by age group and by region. For instance, seasonal patterns among children aged 1 month–14 years differed across all climate regions and two regions (hot semi-arid and tropical savannah) did not exhibit bimodal annual peaks. This heterogeneity strongly implies there was no systematic bias to detecting more deaths in the months of January and July.

9) An explanation is required on some of the reasons on why there is significant reduction (from 77.4% in Table1 to 37.9% in Table 2) in agreement between two the physicians for initial assignment for pneumonia. This has implications on further analysis and reporting that have been conducted for adult age subset, for example average seasonal patterns of pneumonia deaths in Figure 3. An explanation on how this disagreement could have biased the results and how it was resolved would improve the manuscript.

We agree the reduction in physician agreement with increasing age is of interest, though was largely not in the purview of this study. We have updated the Discussion section as follows:

“Notably, there was high agreement (>75%) for child pneumonia and diarrhea deaths while there was poor agreement for adult pneumonia deaths (38%). As such, greater misclassification may occur for adults due to the more disparate pathways of exposure and clinical disagreement resulting from potential comorbidities and further study of adult pneumonia is warranted.”